# Spectrum decomposition in Gaussian scale space for uneven illumination image binarization

**JianWu Long, ZeRan Yan** [ORCID]**\*, HongFa Chen, XinLei Song**

College of Computer Science and Engineering, Chongqing University of Technology, Chongqing, China

\* yan3142537@gmail.com

## Abstract

Although most images in industrial applications have fewer targets and simple image backgrounds, binarization is still a challenging task, and the corresponding results are usually unsatisfactory because of uneven illumination interference. In order to efficiently threshold images with nonuniform illumination, this paper proposes an efficient global binarization algorithm that estimates the inhomogeneous background surface of the original image constructed from the first $k$ leading principal components in the Gaussian scale space (GSS). Then, we use the difference operator to extract the distinct foreground of the original image in which the interference of uneven illumination is effectively eliminated. Finally, the image can be effortlessly binarized by an existing global thresholding algorithm such as the Otsu method. In order to qualitatively and quantitatively verify the segmentation performance of the presented scheme, experiments were performed on a dataset collected from a nonuniform illumination environment. Compared with classical binarization methods, in some metrics, the experimental results demonstrate the effectiveness of the introduced algorithm in providing promising binarization outcomes and low computational costs.

**Data Availability Statement:** All relevant data are within the paper and its Supporting information files.

**Funding:** The Humanities and Social Sciences Research Key Program of Chongqing Municipal

## Introduction

Image segmentation is a fundamental but challenging research subject in computer vision and has been widely applied in several fields such as military, agriculture, medicine, etc. In industrial applications, image segmentation or binarization is a key preprocessing step in document or gear image analysis and recognition, and it requires higher accuracy and lower time costs.

According to the utilization level of semantic image information, we can divide segmentation methods into two parts. First, threshold-based, region-based, and edge-based methods, which are traditional methods, use low-level semantic information [1]. Second, there are also semantic segmentation methods based on high-level semantics. For example, methods based on deep learning extract the semantic information of the object categories contained in image regions [2]. Because of production requirements, most industrial image scenes are single and repetitive, and binarization requires high real-time performance. Hence, deep learning segmentation methods are not suitable for this kind of task on account of the high costs (rich

Education Commission (Grant No. 17SKG136), the National Natural Science Foundation of China for Young Scientists (Grant No. 61502065) and the Foundation and Frontier Research Key Program of Chongqing Science and Technology Commission (Grant No.cstc2015jcyjBX0127).

**Competing interests:** The authors have declared that no competing interests exist.

datasets, labeling images, training models, and model efficiency). Among the binarization methods for industrial images, due to simplicity, efficiency, and comprehensibility, the thresholding technique has been extensively studied and applied. Thresholding, which can be divided into local methods and global methods, calculates one or more thresholds based on the pixel values of one image and compares the gray value of each pixel in the image with the thresholds [1]. For unevenly illuminated images, the contrast between the foreground and background is less intense, which is why a global method may not threshold these images accurately. Therefore, for uneven lighting environments, we choose a local method to divide an image into several regions and select the thresholds or the thresholds at each point are dynamically selected according to a certain neighborhood range.

Thresholding under realistic unevenly illuminated image scenes has been studied by Cheriet et al. [3], who proposed a recursive thresholding method that extended the Otsu algorithm [4] to segment checked images; however, when the targets are not the darkest, such a recursive method cannot extract the targets correctly. Similarly, based on the Otsu method, Chou et al. [5] proposed a region-based image segmentation algorithm, and Wen et al. [6] proposed a curvelet transform method; however, the former was poorly adaptive, and the latter was sensitive to noise. Bhandari et al. [7] suggested that the multilevel 3-D Otsu with spatial context performed better than the histogram-based Otsu method in segmentation. The text image binarization algorithm based on the precipitation model was proposed by Oh et al. [8]. This method needs to extract edge pixels as seed points, and it has poor processing of weak edges and low contrast images. Yang et al. [9] considered that brightness combined with contextual information between pixels can improve segmentation accuracy. Li et al. [10] introduced a nonlinear thresholding method on vessel support regions. The method based on fuzzy sets that introduced different membership functions to divide pixel categories was proposed by Aja-Fernández et al. [11]. They believed that the use of the multiregion thresholding method can reduce the possibility of pixels being misclassified and overcome the situation of images being affected by artifacts and noise. Fuzzy clustering-based methods include Ma et al.'s improved FCM method [12]. Alvarez et al. [13] proposed a thresholding method using a combination of shadow-invariant feature space and model-based classifiers to address the presence of shadows and lighting problems on roads. More studies have focused on adaptive threshold segmentation algorithms. Wei et al. [14] obtained the adaptive threshold segmentation algorithm of multidirectional gray-wave fluctuation transformation. Long et al. [15] proposed the adaptive minimum error thresholding segmentation algorithm based on the precipitation model. Shen et al. [16] introduced the three-dimensional histogram reconstruction method. Long et al. [17] proposed a Gaussian space method to segment nonilluminated text images. In addition, Ntirogiannis et al. [18] proposed the pixel-based evaluation method. Cai et al. [19] combined Otsu and iterative search to form an image subregion thresholding method. Jia et al. [20] used the structure symmetrical pixel (SSP) method to successfully segment degenerate text images. Wang et al. [21] proposed a more robust method for segmenting images of different modes with high accuracy, which can be applied to a variety of images. In addition to these methods, there are other superior methods for segmentation. For instance, the k-means [22] is a classical and effective algorithm in some segmentation scenes, and piecewise flat embedding [23] (PFE) is an edge-based method that uses Laplacian eigenmap embedding with an $L_{1,p}(1 < p \leq 1)$ regularization term.

In this work, we extend Long's work [17], which used a multiscale architecture to extract image features. In segmenting applications, multiscale methods, such as the U-net structure [1], multipath decoding [24] from deep learning and the multipyramid image spatial structure [25] from traditional methods, are frequently utilized to obtain deep image information. However, Long's work is not robust enough in certain aspects, such as parameter insensitivity and

an unsatisfactory anti-interference capability. Thus, the proposed binarization method uses more flexible parameters for experiments, and it has better results, better real-time performance and higher accuracy within a smaller parameter range than Long's work. Therefore, the experiments proved that our method has better robustness and performance.

The remainder of this paper is organized as follows. In the materials and methods section, we illuminate our method step by step. The next section is the experiments and analysis, which provide the quantitative data and demonstrate the visual results, respectively. The final section is the conclusions and perspectives.

## Materials and methods

Under the interference of nonuniform illumination, even a well-understood picture cannot be binarized completely by global thresholding methods. Fig 1 demonstrates that the Otsu method binarizes an earphone image with uneven illumination, and it can be seen that the foreground (earphone) and background are separated unsuccessfully. The reason is that an object illuminated by strong light tends to cause an uneven distribution of image pixels, which deviates from the original pixel value of the object; and the binarization algorithm is difficult to implement on such images. Therefore, in order to binarize the foreground effectively, it is necessary to eliminate uneven illumination.

The proposed method in this paper consists of the following four parts and is illustrated in Fig 2. First, multiscale denoising and Gaussian scale space generation are performed in steps 1 and 2, respectively. Moreover, images in the Gaussian scale space are fused, and the dimension is reduced by spectrum decomposition in step 3. In addition, $\gamma$ correction is used to improve the brightness and contrast of the intermediate result in steps 4 and 5. Finally, in step 6, the final result is obtained by the Otsu thresholding segmentation algorithm. These parts will be individually explained in the following content.

### Multiscale median filtering

In order to efficiently achieve background-foreground separation, as well as reduce the amount of calculation, we only apply the grayscale information of input images. Via observation and statistics, we concluded that the images with nonuniform illumination tend to contain abundant salt-and-pepper noise, Gaussian noise, or more likely mixed noise. We would

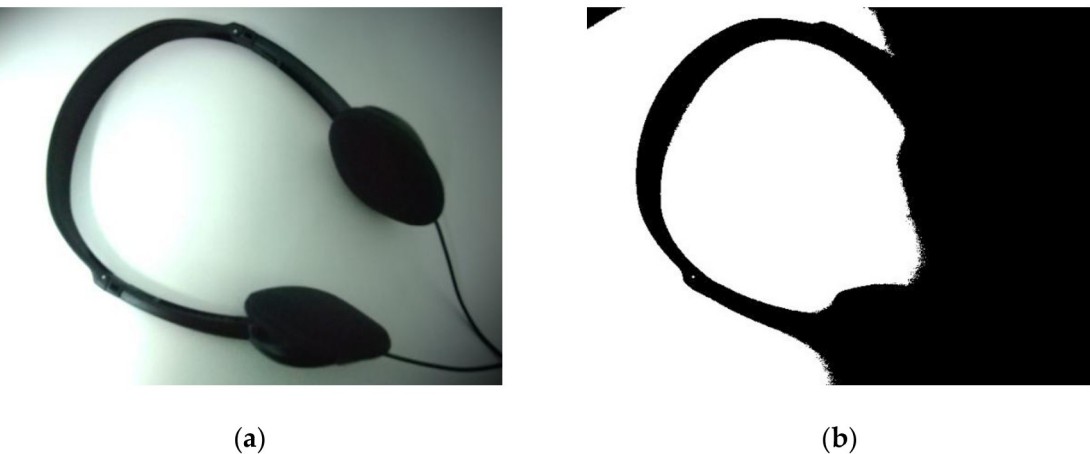

(a)                                             (b)

**Fig 1. Demonstration of (a) Earphone image and (b) Otsu binarization result.** Some methods cannot correctly identify binary images with unevenly illuminated backgrounds.

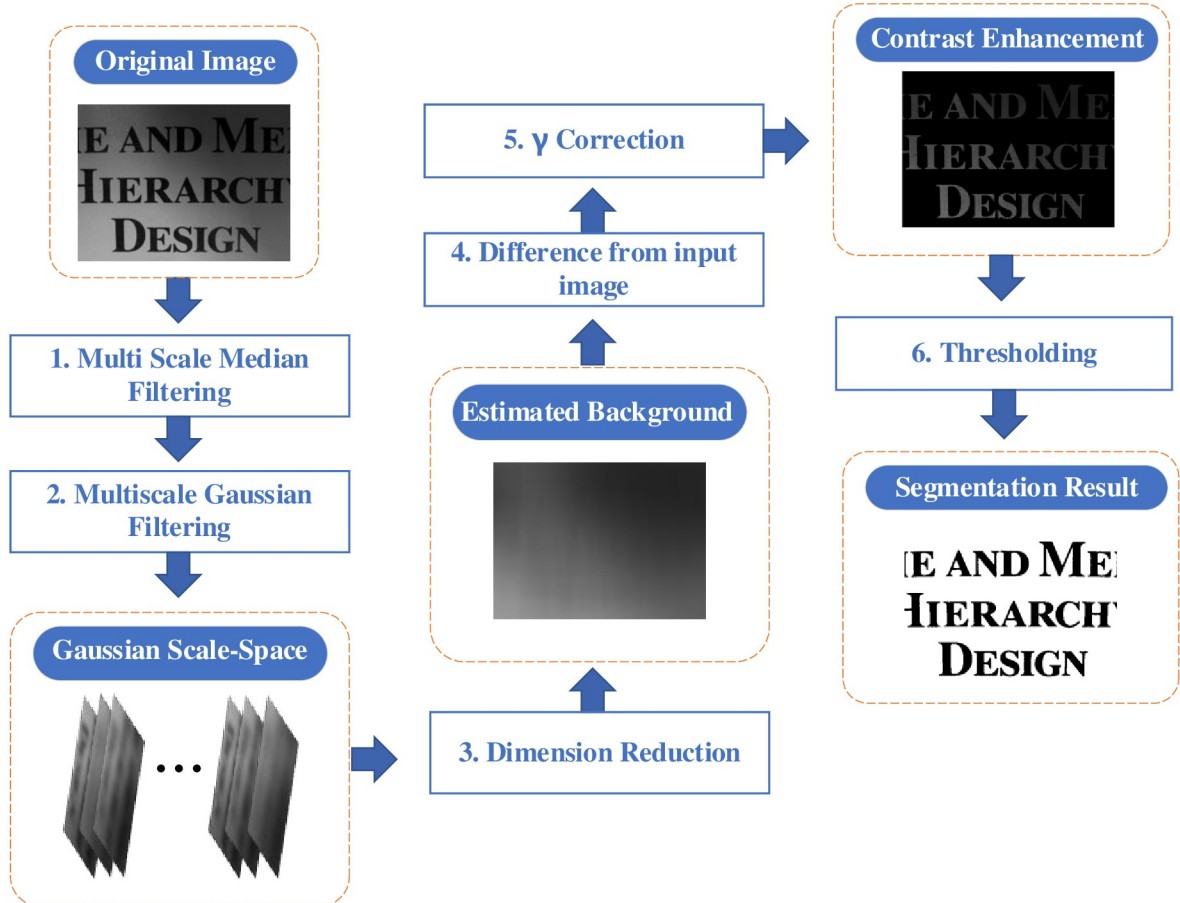

**Fig 2. Flowchart of all processes.** The square rectangles with blue font are the calculation processes, and the round rectangles with white font represent the calculation results.

like to reduce the interference of noise and enhance the anti-noise ability of the method; thus, first, multiscale median filters are used to remove this noise. We use different scale templates for each pixel, and the length and width of the template are $h$ and $w$, respectively, where $w = h = 2n + 1$, $n \in N$. We formulate the two-dimensional median filtering and image operation as follows:

$$I_n(x, y) = median\{I(x + i, y + j)\} \tag{1}$$

where $i = 0,1,2,\ldots, w - 1$; $j = 0,1,2,\ldots, h - 1$; $I(x, y)$ is the input image; and $I_n(x, y)$ is the corresponding filtered result.

Therefore, as $h$ and $w$ increase, image denoising proceeds repeatedly, and these denoised results compose a multiscale space of the median filter image:

$$I_N = \{I_0, I_1, I_2, \ldots, I_n\} \tag{2}$$

## Gaussian scale space

The Gaussian function plays a pivotal role in many fields. In image processing, the two-dimensional Gaussian function is often normalized as a given template with fixed weights. Weights

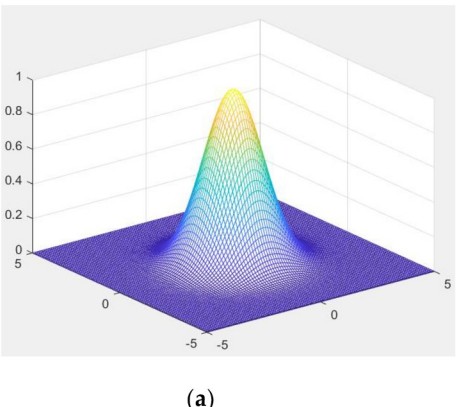

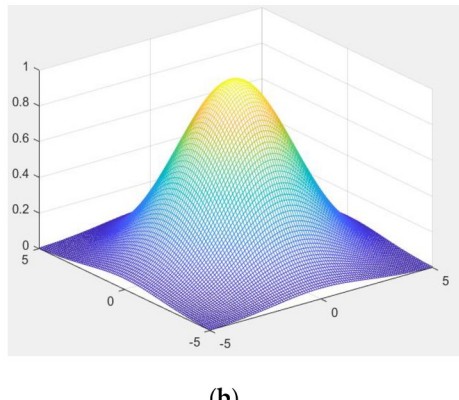

(a) (b)

**Fig 3. Gaussian function surface with different σs.** (a) σ = 1.5, and (b) σ = 3.

are only subject to parameter σ, which means that templates of different scales correspond to different scales. The equation is shown as follows:

$$G(x, y, \sigma) = \frac{1}{2\pi\sigma^2} e^{-\frac{x^2+y^2}{2\sigma^2}} \tag{3}$$

Fig 3 is the visual surface of the two-dimensional Gaussian function, where σ = 1.5 (Fig 3a) and σ = 3 (Fig 3b). Fig 3 reveals that the Gaussian function's use in computer vision is that the weight near the center point has the largest weight; and the farther away a point is from the center point along the circumference, the smaller the weight. This highlights the importance of the center pixel point.

Therefore, because of the above characteristics of the two-dimensional Gaussian filter, the background estimation method used in this paper is completed by a multiscale Gaussian filter function. This type of multiscale space is Gaussian Scale Space (GSS). Furthermore, in the process of creating a Gaussian scale space, the Gaussian filter can relieve the interference of Gaussian noise.

After obtaining $I_N$ in Eq 2, we can establish a Gaussian scale space (GSS) for each image $I_t$. The first image in $I_N$ is $I_0$, and the operation of the Gaussian function and $I_0$ is given as the following equation:

$$L_{0,0}(x, y, \sigma_0) = G(x, y, \sigma_0) * I_0(x, y) \tag{4}$$

where $*$ is the convolution operator. $\sigma_0$ is the first level of the Gaussian scale factor, and it is initialized empirically by researchers. The image obtained by the Gaussian filtering operation for each subsequent layer can be expressed by the following equation:

$$L_{i,j}(x, y, \sigma_j) = G(x, y, \sigma_j) * I_i(x, y) \tag{5}$$

where $L_{i,j}(x, y, \sigma_j)$ represents the output of the $j^{th}$ layer of $I_t$ in the GSS.

There should be a large difference among the scales σ so that the Gaussian function can effectively extract the background feature from unevenly illuminated images. In order to obtain multiscale σ, we can calculate σ using the following equation:

$$\sigma_j = \sqrt{\beta}^j * \alpha \tag{6}$$

where $\beta = 0.1, 0.2,\ldots, n$; $\alpha = 1, 2,\ldots, m$; scale $\sigma_j$ is determined by the hyperparameters $\alpha$ and $\beta$; and $j$ is the number of iterations of the Gaussian scale.

We can establish each GSS using the above scheme. However, if $\sigma$ is too large, all pixels will become a certain value, which will deteriorate the extraction of the image background. In order to prevent this situation from happening, we use $\Delta_j$ to control the establishment of the GSS, where $\Delta_j$ is the average pixel difference between the images of two adjacent layers of Gaussian filtering results:

$$\Delta_j = \frac{1}{W \times H} \sum_{x=0}^{W-1} \sum_{y=0}^{H-1} |L_j(x, y, \sigma_j) - L_{j-1}(x, y, \sigma_{j-1})| \tag{7}$$

where the image size is $W \times H$. When $\Delta_j$ is less than a given threshold $\varepsilon$, the establishment of the GSS is stopped. In this process, each image $I_t$ is smoothed with the exponential calculation of $\sigma$ so that each layer output image $L_{i,j}(x, y, \sigma_j)$ is gradually approaching the background, which also makes $\Delta_j$ decrease gradually. After the error threshold $\varepsilon$ is given, when $\Delta_j < \varepsilon$, it means that the extracted image no longer has more effective features and information. At this time, we can stop establishing the GSS.

How we build the whole GSS is demonstrated in Fig 4, where the horizontal axis is the median filter result $I_N$, $w$ is the scale of the median filter template, and the vertical axis is the establishment of Gaussian scale space. We know that when the scale of the Gaussian filter template is smaller, more prominent target boundary information can be extracted. At this time, the target in the extracted image has complete local features, such as edge details and bright color, but the method was not effective at extracting background information. When $\sigma$ gradually increases, the ability to extract the overall information of the target is enhanced. For

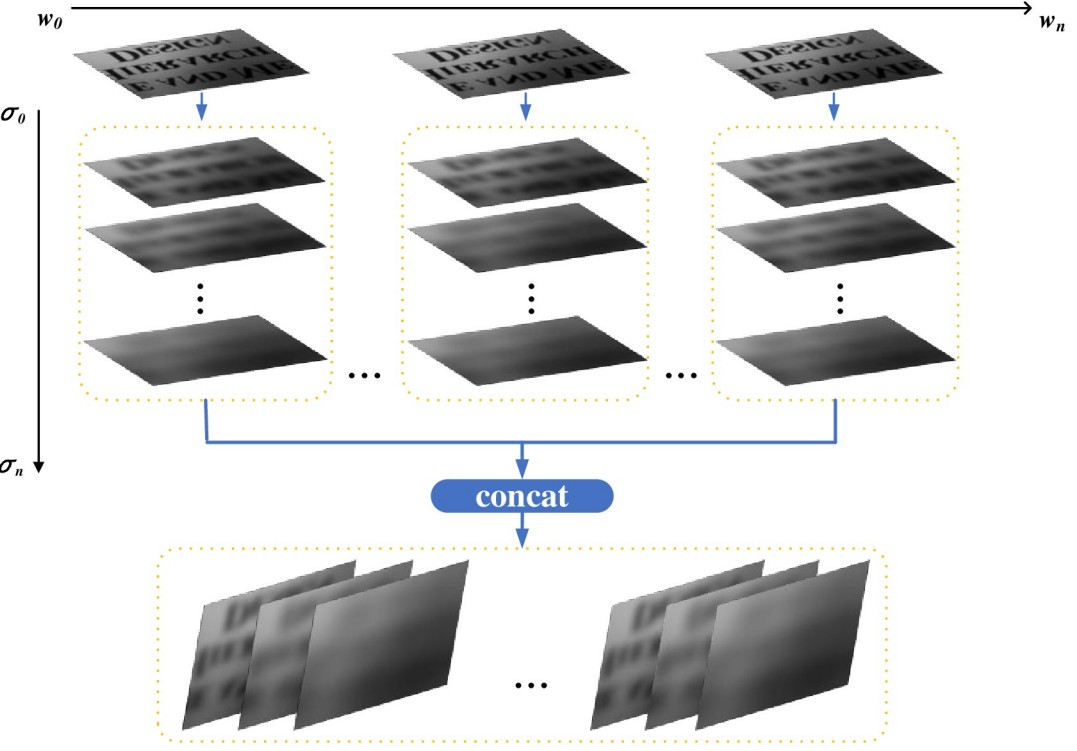

**Fig 4. Establishment of the GSS.**

instance, the background can be better extracted, and the relationship between the object and the background is better represented.

## Nonuniform illumination background extraction

After obtaining the GSS, we need to estimate the nonuniform illumination background. We choose PCA (Principal Component Analysis) for dimensionality reduction to determine the uneven illumination. PCA, in which the derived uneven illumination is a linear representation of the GSS, is widely used in many disciplines and fields. Based on what we described in the above content, the equation $L = \{L_0, L_1, L_2, \cdots, L_N\}$, where $L_i \in R^{w \times h}$ ($i = 0,1,\ldots,N$), and $w$ and $h$ are the width and height of the input picture, respectively, can be obtained.

Therefore, the PCA algorithm is used to calculate the principal component of stacked $L$s and to facilitate the calculations. We convert the matrix of $L$ to training data $X = (x_0, x_1,\ldots, x_N)^T$, where $x_i \in R^M$ ($i = 0,1,\ldots,N$) denotes a vector flattened by $L_i$, and $M = w \times h$.

The PCA algorithm finds a set of bases that comes from training data $X$ with zero mean so that $X$ can be mapped into a linear space composed of this set of bases. The purpose is to achieve data reduction and feature extraction. This set of bases $V$ is obtained from the spectrum or eigendecomposition of the covariance matrix $C$ of the data $X$, where $C$ is formulated as:

$$C = \frac{1}{N}X^T X \tag{8}$$

The spectrum decomposition result consists of an eigenvector matrix $G$ and an eigenvalue matrix $K$:

$$C = GKG^T \tag{9}$$

Because $C$ is very large and has high dimensionality, directly performing the eigendecomposition of the covariance matrix $C$ requires a large computational cost. Therefore, the better choice is that we apply SVD to X as follows:

$$X = U\Sigma V^T \tag{10}$$

where $U \in R^{N \times N}$, $\Sigma \in R^{N \times M}$, $V \in R^{M \times M}$, $VV^T = E_M$, $UU^T = E_N$, $U$ and $V$ are matrixes composed of a standard orthogonal basis, and $E_M$ and $E_N$ are unit matrixes with dimensions $M \times M$ and $N \times N$, respectively. Then, the covariance matrix $C$ can be rewritten as:

$$C = V\Sigma^2 V^T \tag{11}$$

Comparing Eq 11 with Eq 9, we find that $V = G$ and $\Sigma^2 = K$. Hence, we obtain $V'$, which is the first $k$ columns in $V$; and then obtain a set of coordinates $P$ by mapping $X$ to $V'$:

$$P = XV' \tag{12}$$

where $V' \in R^{M \times k}$ and $P \in R^{N \times k}$.

Finally, we can reconstruct the image data denoted as $X_{predict}$ by applying the linear transformation matrix $V'$ to the low-dimensional coordinates P as

$$X_{predict} = P(V')^T \tag{13}$$

where $X_{predict} \in R^{N \times M}$. In addition, for $X_{predict}$, we use the first principal component in this work, which is defined as $X_{predict}$, for the subsequent processing.

## Image gamma correction

Gamma ($\gamma$) correction is an operator that is used to perform nonlinear operations on image signals. As a result, it is an operation used to adjust the light and dark range of image gray values. After the estimated background image is obtained and the interference of uneven background is eliminated by using a difference method, the obtained target image will appear relatively dark. $\gamma$ correction can be used to highlight the contrast between light and dark, making the target more prominent.

$$f(x) = cx_{predict}^{\gamma} \tag{14}$$

where $c$ is a constant, $\gamma$ is the scale of the nonlinear transformation, and $x_{predict}$ comes from Eq 13.

When $\gamma$ is less than 1, the lower gray level regions in the image will be stretched, and the higher gray level portions will be compressed. In this way, the details of the dark parts of the image are improved. When $\gamma$ is greater than 1, the areas with higher gray levels in the image will be stretched, and the parts with lower gray levels will be compressed; therefore, the contrast of the image will be significantly enhanced.

After applying the difference method to the estimated background image with the original image, we can obtain a target image, but the result at this time is relatively dark, as shown in Fig 5. Therefore, gamma correction can improve the brightness and obtain the prominent target in Fig 6, where $\gamma = 0.4$.

## Thresholding

The last procedure of our method is thresholding. After we obtain gamma-corrected images, the classic Otsu method is used to perform binarization. Otsu is a global thresholding method proposed by the Japanese scholar Nobuyuki Otsu in 1980. The algorithm calculates the histogram of the image, uses the maximum variance of the background and target as the threshold selection criteria, and then uses the threshold to binarize the image.

First, the interclass variance $g$ is obtained from the image. In Eq 15, the proportion of pixels belonging to the foreground relative to the entire image is recorded as $\omega_0$, and its average grayscale is $\mu_0$; furthermore, the proportion of background pixels relative to the entire image is $\omega_1$,

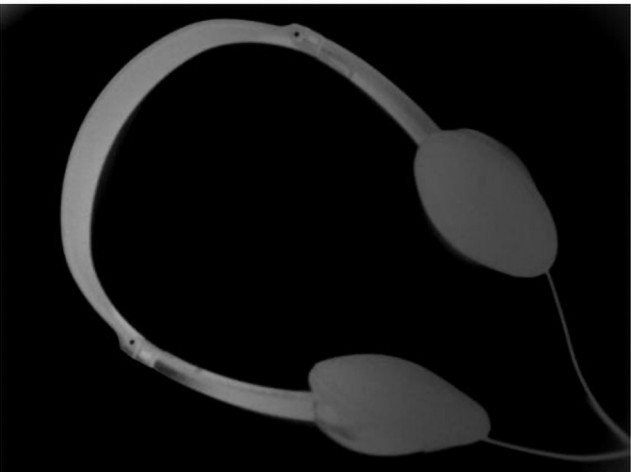

**Fig 5. Estimated target image.**

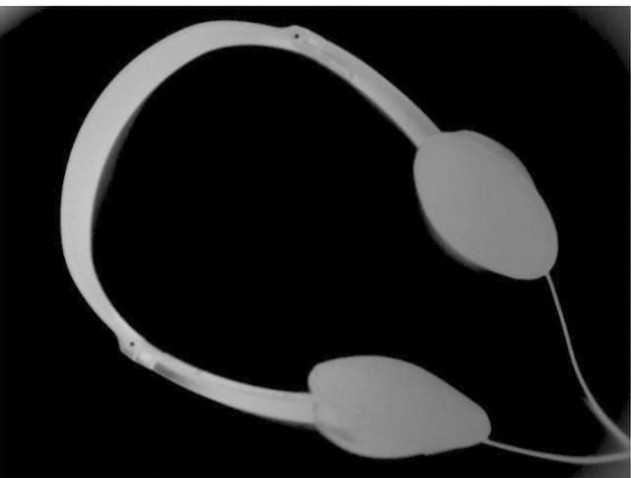

**Fig 6. Result of $\gamma$ correction.**

and its average grayscale is $\mu_1$.

$$g = \omega_0 \omega_1 (\mu_0 - \mu_1)^2 \tag{15}$$

Then, all pixels of the image are traversed to find the value with the smallest variance between pixels, and the pixel value is the threshold. Represented as the peak in the image on the histogram, the threshold obtained by the Otsu method is approximately equal to the lowest valley between the peaks. This method is simple and fast, and it is very suitable for the images to be processed in the proposed method.

## Experiments and analysis

This section demonstrates our experiments and analyzes how to choose parameters through experiments. We mainly use the ME (Misclassification Error) as the evaluation standard for the binarization of unevenly illuminated images. The ME denotes the relationship between the prediction and the ground truth (or mask). In other words, the ME is the proportion of the background that is misclassified as foreground or the proportion of the foreground that is misclassified as background.

$$\text{ME} = 1 - \frac{|B_M \cap B_P| + |F_M \cap F_P|}{|B_M| + |F_M|} \tag{16}$$

where $B_M$ and $F_M$ are the background and foreground in the mask, respectively. $B_P$ and $F_P$ are the background and foreground of the binarized image, respectively. The ME measures the difference in the pixels between the predicted segmentation result and the mask image. A lower ME indicates a smaller degree of difference, and the closer the segmentation result is to the standard mask.

In addition, we use the PA (pixel accuracy), MIOU (mean intersection over union), FM (F-measure) and time (running time) as evaluation metrics:

$$PA = \frac{|F_M \cap F_P|}{|B_M| + |F_M|} \tag{17}$$

$$MIOU = \frac{1}{2} \times \left( \frac{|F_M \cap F_P|}{|F_M \cup F_P|} + \frac{|B_M \cap B_P|}{|B_M \cup B_P|} \right) \tag{18}$$

$$FM = \frac{2 \times Recall \times Precision}{Recall + Precision} \tag{19}$$

where recall and precision are the recall rate and precision rate, respectively. They are calculated by Eqs 20 and 21, where TP (true positives) is the proportion of positive samples that are classified as positive samples, TN (true negatives) is the proportion of negative samples that are classified as negative, FP (false positives) is the proportion of negative samples that are classified as positive, and FN (false negatives) is the proportion of positive samples that are classified as negative.

$$Recall = \frac{TP}{TP + FN} \tag{20}$$

$$Precision = \frac{TP}{TP + FP} \tag{21}$$

In order to verify the feasibility of our method, we compared it with the Otsu algorithm [4], the water flow algorithm [8], Wang's algorithm [21], the SSP [20], the k-means algorithm [22] and PFE [24]. The comparison methods used the parameter ranges recommended by the authors in the respective papers in the experiments. Finally, the experiments illustrate that the method proposed in this paper has better performance and robustness than these comparison methods in nonuniformly illuminated image binarization, regardless of the visual aspect or evaluation. All experiments were completed on the same computer, which includes an Intel I7-7700 CPU, an HP 8298 motherboard, 16 GB of memory, and the Windows 10 64-bit operating system.

Our method is determined by the hyperparameters α, γ, and β. In the first step, we calculate the distribution of the four evaluation indicators when the hyperparameters range from 10 to 99 for α, from 0.1 to 0.9 for γ, and from 0.5 to 2.4 for β on a dataset including hundreds of images that was collected from a nonuniform illumination environment. Furthermore, we determine the distribution to find the parameter ranges that result in the best performance.

## Text image experiment under uneven illumination

This section segments text images with uneven illumination. We tested 8 text images with different scales and lighting environments. The images and masks are shown in Figs 7 and 8 below, where Fig 7 includes brighter images and Fig 8 includes darker images.

The figures show that in these images, the targets are exposed to different extents of lighting, which are distributed differently. Therefore, the granularity of the lighting of each image is different.

After many experiments seeking to determine the hyperparameter ranges, we conclude that when α ranges from 10 to 99, γ ranges from 0.1 to 0.9, and β ranges from 0.5 to 2.4, the method

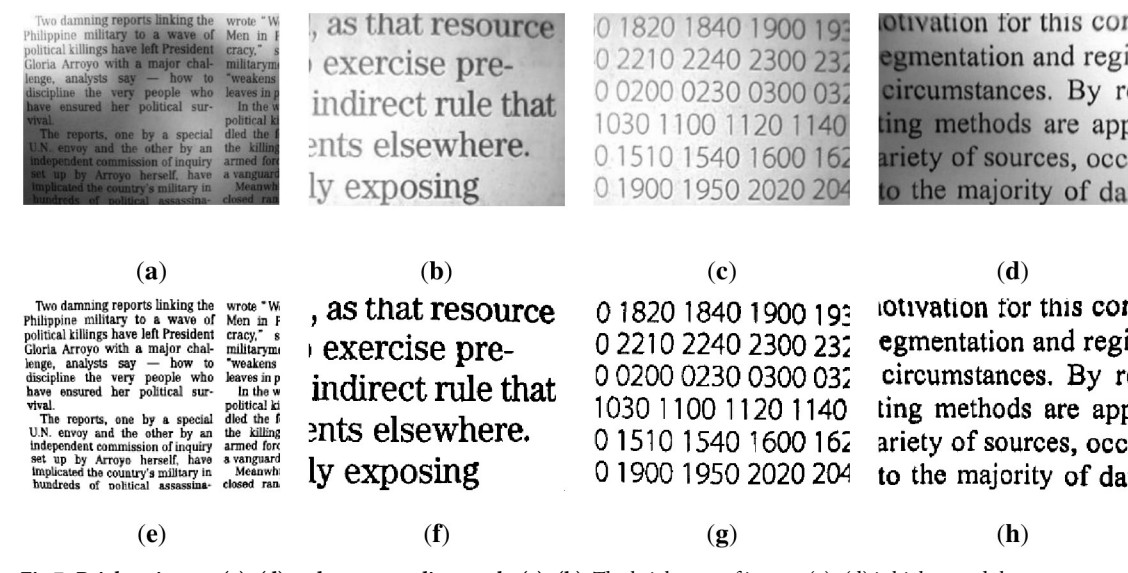

**Fig 7. Brighter images (a)~(d) and corresponding masks (e)~(h).** The brightness of images (a)~(d) is higher, and the targets are clear. In addition, images (e)~(h) are the result of manual segmentation corresponding to (a)~(d).

performed well on all text images; and the average evaluation results are ME: 0.056, MIOU: 0.8514, PA: 0.9439, F_Measure: 0.9641, and time (ms): 596.

**Parameter selection.** The running time distribution is shown in Fig 9 below. It shows that when $\alpha < 20$, $\beta < 1.0$ and $0.1 \leq \gamma \leq 0.9$, our method costs more time. However, when $\alpha$ and $\beta$ are within the appropriate ranges, the timing costs are absolutely low, and all are below 2 seconds.

To find parameters with satisfactory binarization performance, we first assess $\gamma$ when the ranges of $\alpha$ and $\beta$ are from 10 to 99 and 0.5 to 2.4, respectively. Fig 10a below shows the curve of the average ME when $\gamma$ ranges from 0.1 to 0.9; and Fig 10b shows the curve of the average MIOU, PA, and F_Measure when $\gamma$ ranges from 0.1 to 0.9. The figures show that the best $\gamma$ is between 0.3 and 0.5.

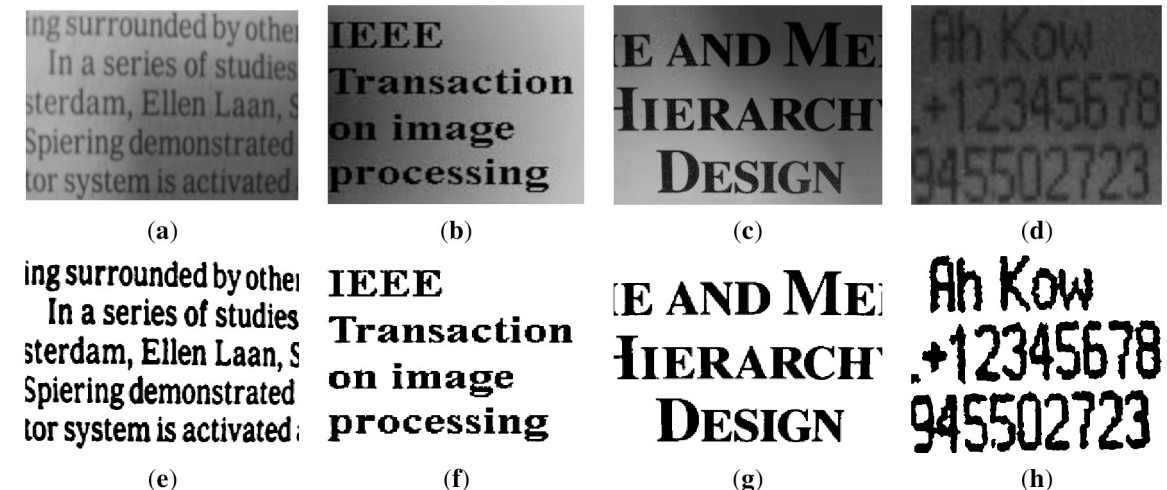

**Fig 8. Darker images (a)~(d) and corresponding masks (e)~(h).** The brightness of images (a)~(d) is lower, and the targets are slightly blurred. In addition, images (e)~(h) are the result of manual segmentation corresponding to (a)~(d).

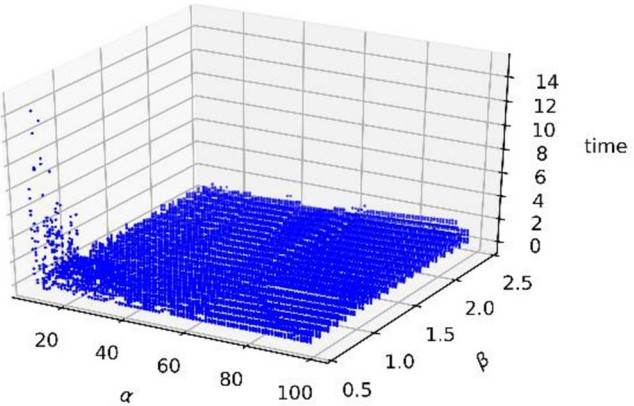

**Fig 9. Scatter distribution of the running time.**

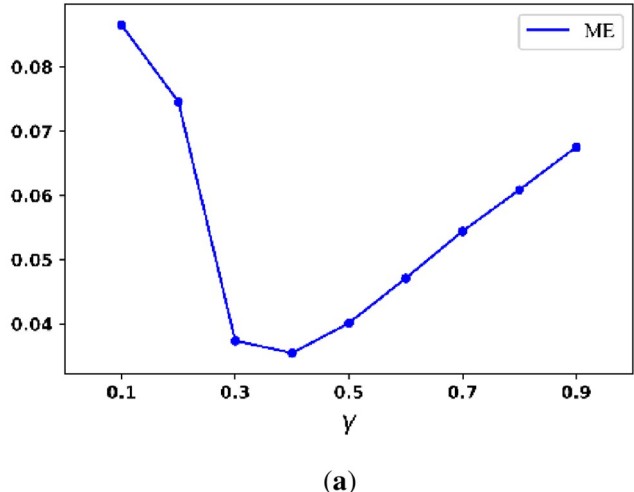

(a)

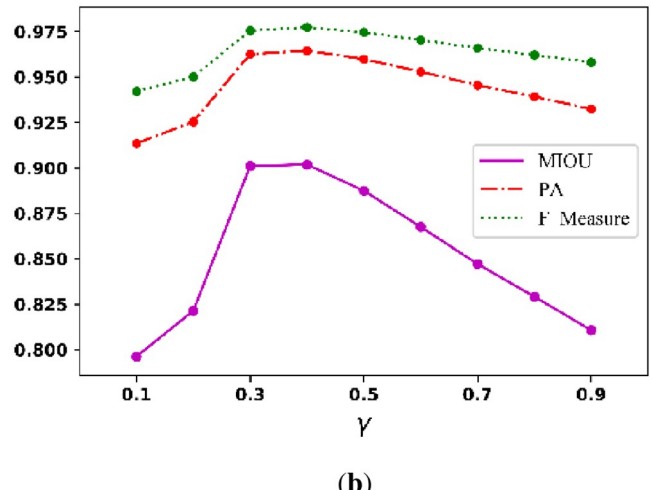

(b)

**Fig 10. Average (a) ME curve and (b) MIOU, PA, F_Measure curves.**

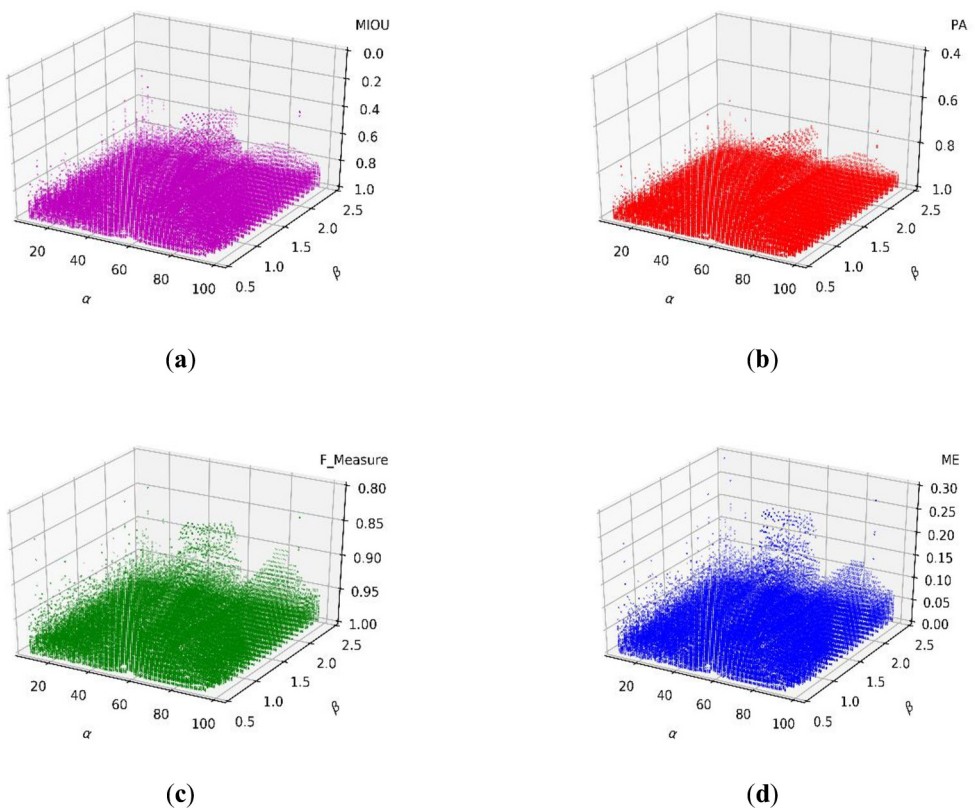

**Fig 11. Scatter distributions of the (a) MIOU, (b) PA, (c) F_Measure, and (d) ME.**

As a result of the experiments that can determine γ above, we fixed γ at 0.4 and assessed the ranges of α and β. Fig 11a–11d show the scatter distributions of the MIOU, PA, F_Measure, and ME, respectively.

The experimental results in Fig 11 show the statistical distributions of the three parameters. Therefore, we can define that the ranges that result in the best performance are when α ranges from 10 to 50, β ranges from 0.8 to 2.2, and γ ranges from 0.3 to 0.5.

**Text image results and comparison.** According to the previous chapter, which has found the ideal parameters for experiments, in this chapter, we tested and compared several nonuniformly illuminated text images from the dataset based on the previous experiments. Each image has a different size and is exposed to different amounts of light and lighting positions. In order to prove the effectiveness of our method for binarizing the uneven illumination of images, we used this dataset as experiments can yield fair and convincing results.

First, we reveal the segmentation results of the brighter image from Fig 7a in Fig 12. The results show that the water flow method (Fig 12a) does not clearly illustrate the punctuation of the text, and the background is classified as foreground (marked by a red frame). Fig 12b shows that the SSP method performs very well overall, but there are some cases of adhesion on the text, which means that some text segmentation is not very clear. The remaining results of the comparison methods are shown in Fig 12c–12e, which are unacceptable image segmentation results when there are many unevenly illuminated areas in an image.

In addition, we demonstrate the segmentation results of the image from Fig 8d in Fig 13, where Fig 8d is an uneven image on a darker background. It can be seen that the segmentation results of some comparison methods such as the water flow method, Wang's method, Otsu

**Fig 12. The results of Fig 7a binarized by several methods.** (**a**) water flow method, (**b**) SSP method, (**c**) Wang's method, (**d**) Otsu method, (**e**) k-means method, and (**f**) proposed method. Some flaws of the comparison methods are shown in red frames.

method and k-means method are not satisfactory for this type of image because the text objects in uneven illumination are not well extracted. The SSP method worked ideally; however, it also had some cases of adhesion, which resulted in lower effectiveness for small targets, such as punctuation marks, and it easily segmented some targets (text) as background by mistake.

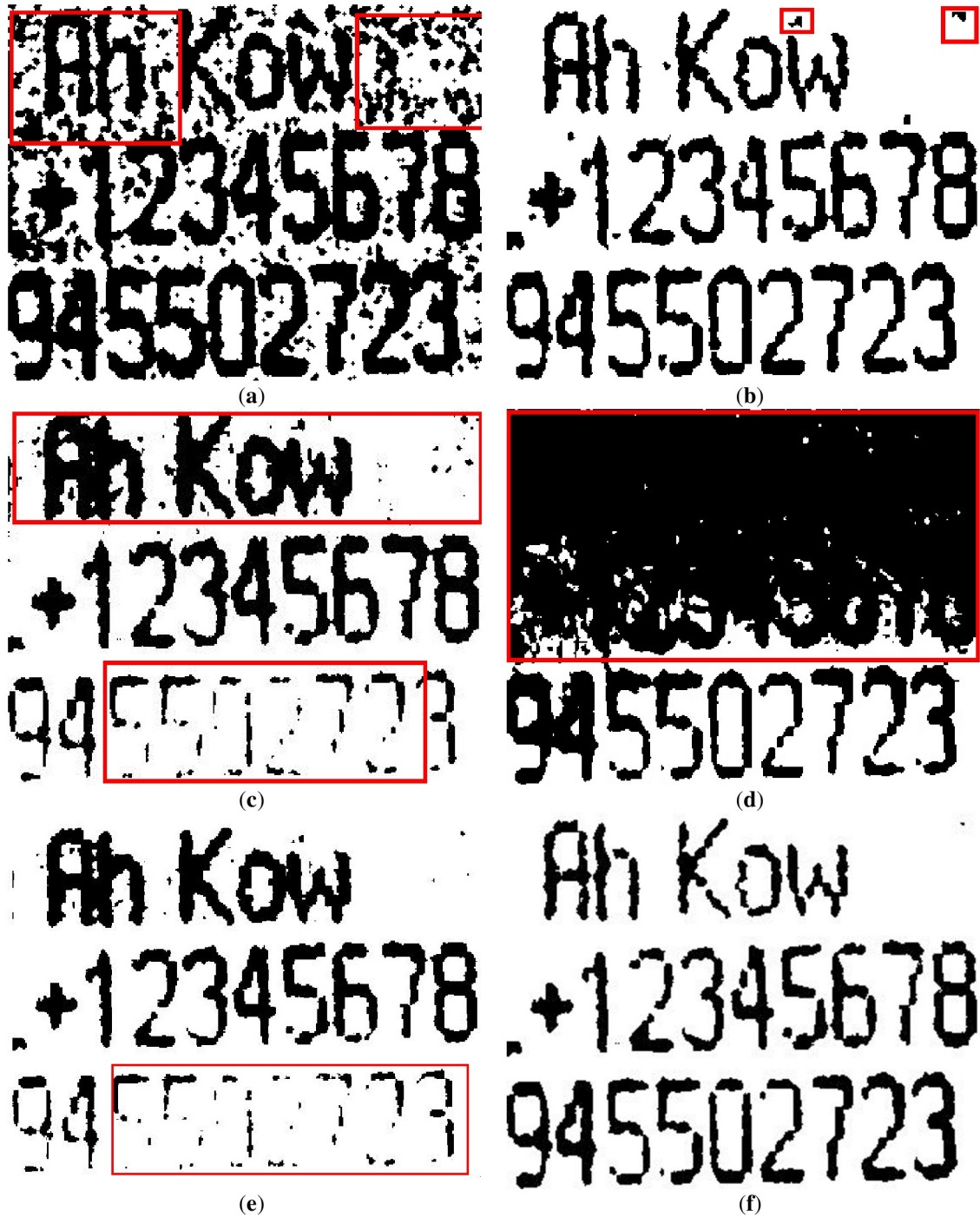

**Fig 13. The results of Fig 8a binarized by several methods.** (**a**) water flow method, (**b**) SSP method, (**c**) Wang's method, (**d**) Otsu method, (**e**) k-means method, and (**f**) proposed method. Some flaws of the comparison methods are shown in red frames.

In summary, the binarization results of the comparison methods show that the water flow method has better segmentation results for brighter scenes, but its segmentation in lower and uneven illumination environments is not satisfactory. We show other results in Figs 14 and 15. The SSP method performed well, whether in the ME or other evaluation metrics, but this

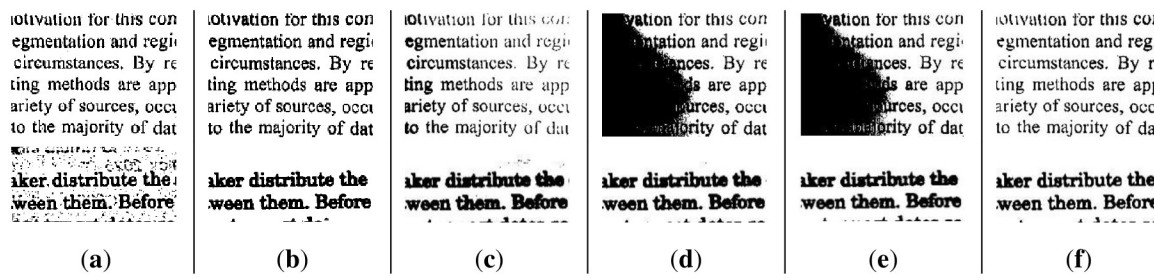

**Fig 14. The results of some text images.** (**a**) water flow method, (**b**) SSP method, (**c**) Wang's method, (**d**) Otsu method, (**e**) k-means method, and (**f**) proposed method.

approach has one small drawback in that the running time costs are slightly higher. Since Wang's method is a global optimal solution for segmentation, it cannot complete the binarization task in images with local uneven illumination, and its evaluation performance was poor. Otsu is a classic algorithm that requires almost no time to run, but its performance is not satisfactory. The k-means method is not suitable for the segmentation of unevenly illuminated text images in terms of most effects.

As for PFE, some text images failed to be segmented because the boundaries of the images in the dataset could not be extracted correctly. Table 1 presents the quantitative comparison results of several comparative methods and the proposed method. The time of PFE is not

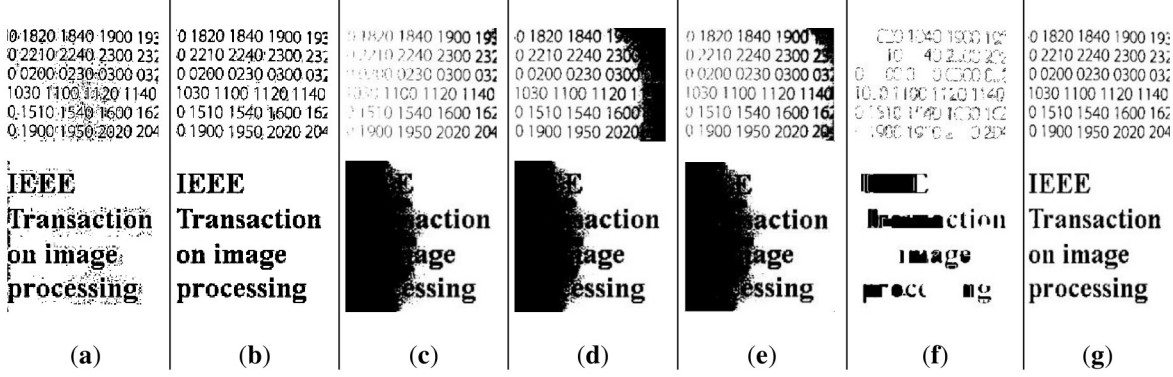

**Fig 15. The results of some text images.** (**a**) water flow method, (**b**) SSP method, (**c**) Wang's method, (**d**) Otsu method, (**e**) k-means method, (**e**) PFE method and (**g**) proposed method.

**Table 1. Quantitative performance of the six comparison methods and our method.**

| Method | ME | MIOU | PA | F_Measure | Time (ms) |
|---|---|---|---|---|---|
| Water flow | 0.0631 | 0.8481 | 0.9368 | 0.9555 | 269 |
| SSP | 0.0353 | 0.9030 | 0.9646 | 0.9772 | 1399 |
| Wang's | 0.3469 | 0.4527 | 0.6530 | 0.7609 | 754 |
| Otsu | 0.2744 | 0.5476 | 0.7255 | 0.7780 | **1** |
| K-means | 0.4160 | 0.3518 | 0.5839 | 0.6820 | 1582 |
| PFE | 0.2008 | 0.5039 | 0.1303 | 0.7796 | - |
| Ours | **0.0312** | **0.9134** | **0.9688** | **0.9799** | 417 |

**Table 2. Quantitative performance of Long's method and ours.**

| Method | ME | MIOU | PA | F_Measure | Time (ms) |
|---|---|---|---|---|---|
| Long's | **0.0173** | **0.9536** | 0.9827 | 0.9449 | 504 |
| Ours | 0.0211 | 0.9405 | **0.9848** | **0.9862** | **351** |

given because it costs minutes for the calculation, and we do not record it for comparison. Combined with visual demonstration and quantitative indicators, we can simply conclude that in the binarization results of text images with nonuniform illumination, some methods are suitable for binarized brighter images while others are suitable for binarization darker images. Our method is suitable for both cases and has good robustness.

Because our method expands on Long's method [17], the comparison results and tables of the two are separately listed to show that our method is more robust and advanced. We also use statistical methods to further narrow the range of parameter values to achieve more accurate binarization; and the experimental results show that $\alpha = 15$, $\beta = 1.2$, and $\gamma = 0.3$ are optimal. The parameters for Long's method are $\sigma = 20$ and $\gamma = 0.3$. It can be seen from Table 2 that the difference in the average performance of evaluation is not prominent, but the method in this paper required less time than Long's method.

However, Long's method performs poorly on binarized images with insufficient light, and it did not remove the background with uneven lighting well. As illustrated in Fig 16, the performance of Long's method is slightly worse in darker environments, such as Fig 16a, which is the binarization result of Fig 8d.

## Nontext image experiments under uneven illumination

The following images are nontext images and masks. The foregrounds of these images are different objects, which have different shapes, appearances, and lighting environments. The images are shown in Fig 17. The first row is the original images, and the second row is the masks.

After rough experiments, we concluded that when $\alpha$ ranges from 10 and 99, $\beta$ ranges from 0.1 to 0.9, and $\gamma$ ranges from 0.5 to 2.4, the average experimental evaluation metrics of all

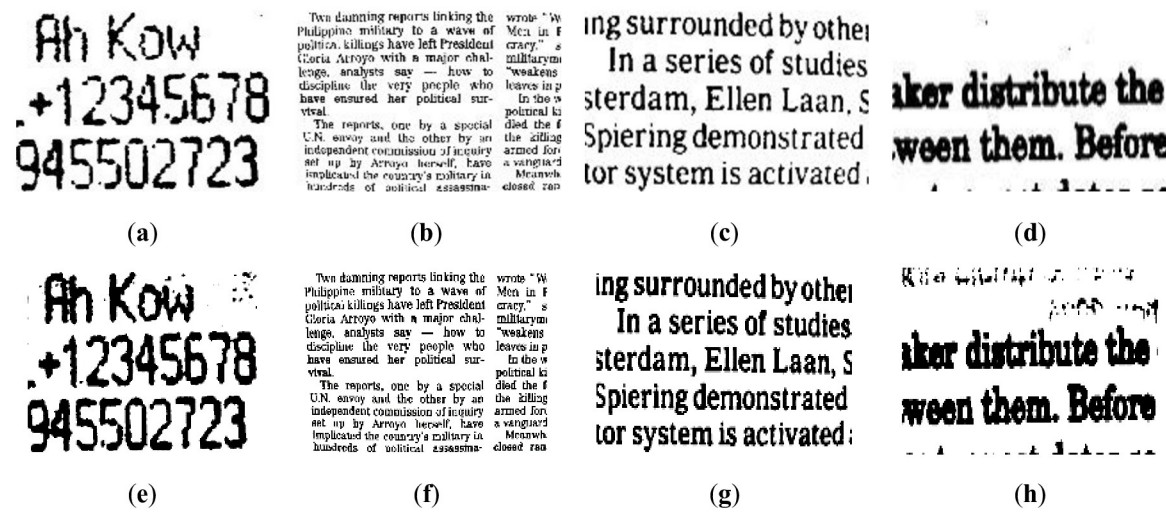

**Fig 16. Demonstration of our method (a)~(d) and Long's method (e)~(h).**

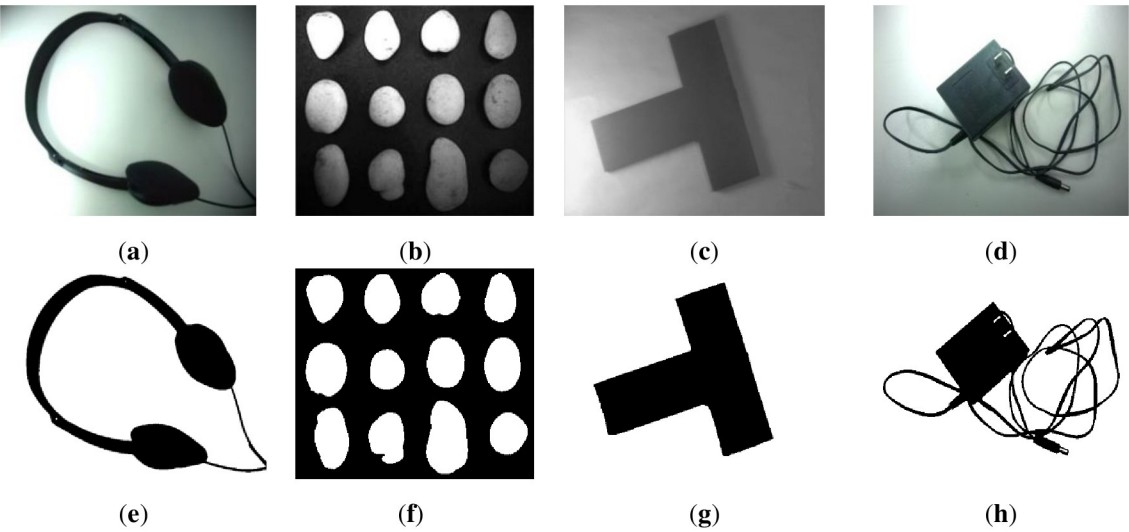

**Fig 17. Nontext images (a)~(d) and their corresponding masks (e)~(h).**

nontext images are ME: 0.0555, MIOU: 0.8742, PA: 0.9444, F_Measure: 0.9538, and time (ms): 680.

**Parameter selection.** Similarly, this subsection uses the same method to determine the three parameters. The running time distribution is shown in Fig 18. It can be illustrated that when α<20 and β<1.0, our method costs more time, which results from consuming considerable time in the construction of the GSS. However, when α and β are within the remaining parameters, because the establishment of the GSS can be completed quickly and the calculation costs of the other steps is low, the timing costs are absolutely low, all below 2 seconds.

To find the parameters that can generate satisfactory binarization performance, we set α and β in the ranges of 10 to 99 and 0.5 to 2.4, respectively. Fig 19a illustrates the curve of the average ME when γ is set from 0.1 to 0.9; and Fig 18b illustrates the curves of the average MIOU, average PA, and average F_Measure when γ is set from 0.1 to 0.9. The figure shows the best setting for γ ranges from 0.3 to 0.5.

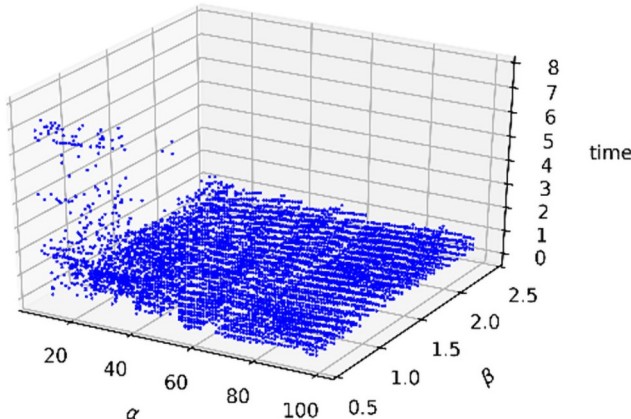

**Fig 18. Scatter distribution of running times.**

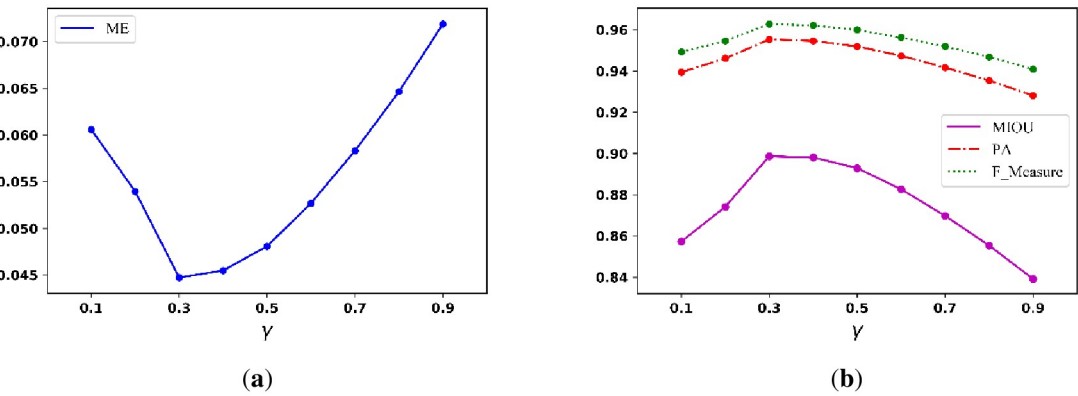

**Fig 19. Average (a) ME curve and (b) MIOU, PA, and F_Measure curves.**

The scatter distributions of the MIOU, PA, F_Measure, and ME are shown in Fig 20a–20d, respectively. We set γ to 0.4 and assessed the value ranges of α and β.

The above experimental results provide a statistical distribution of three parameters. Therefore, we find that the best performance range of α is from 10 to 50, that of β is from 0.8 to 2.2, and that of γ is from 0.3 to 0.5.

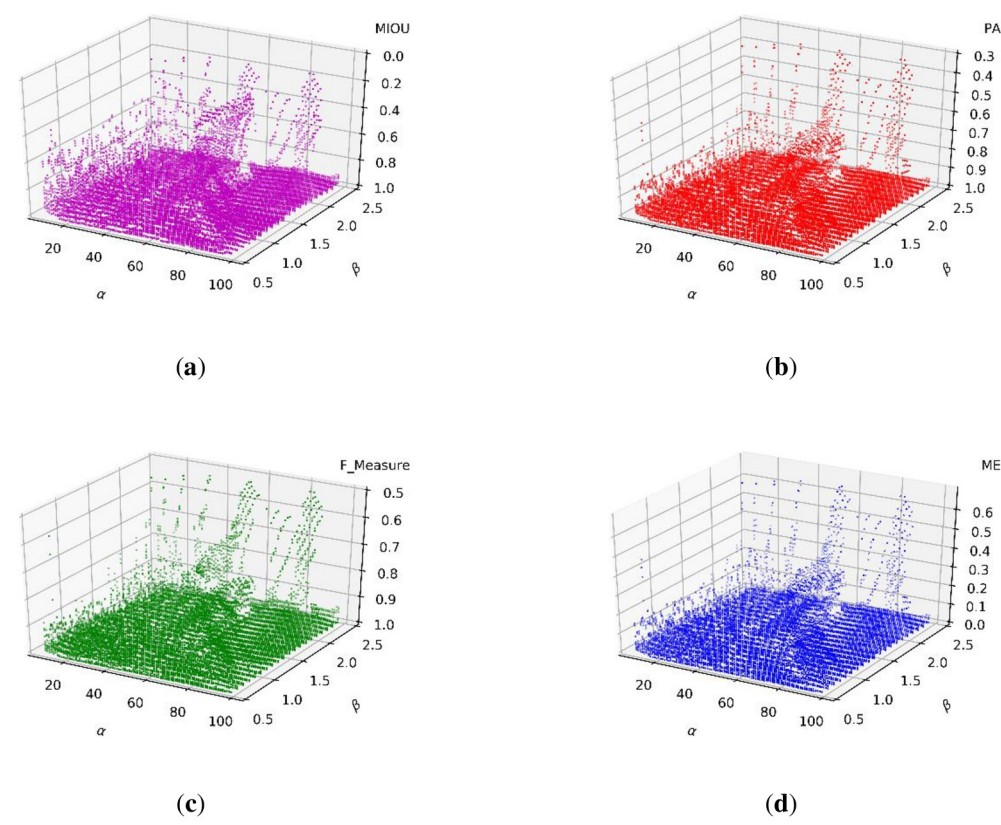

**Fig 20. Scatter distributions of (a) MIOU, (b) PA, (c) F_Measure, and (d) ME.**

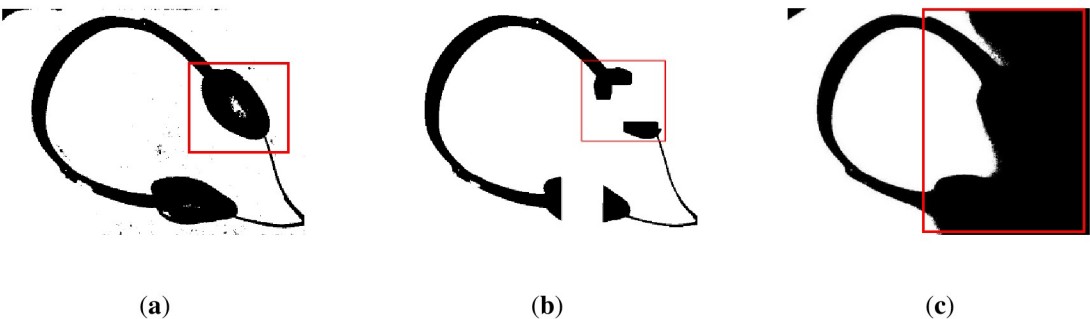

|     (a)     |     (b)     |     (c)     |

**Fig 21. The results of Fig 17a.** (**a**) water flow method, (**b**) SSP method, and (**c**) Wang's method. Some flaws of the comparison methods are shown in red frames.

**Nontext image results and comparison.** In this subsection, we tested and compared all nontext images with inhomogeneous illumination in the dataset based on the above parameter selection experiments. Each image has a different size and is exposed to different amounts of light and lighting positions. We demonstrated the representative segmentation results in images with nonuniform illumination and listed the final evaluation in the table.

We reveal the segmentation results of the image in Fig 17a in Figs 21 and 22. It can be seen that the water flow method (Fig 21a) and SSP method (Fig 21b) cannot completely overcome the influence of uneven illumination; thus, they binarized the image partially. However, the PFE method (Fig 22c) binarized out most of the targets.

The results of other nontext images under inhomogeneous illumination are demonstrated in Figs 23 and 24. The binarized results of Wang's method, Otsu's method and the k-means method are revealed in Figs 23c and 24a and 24b, respectively, which illustrate that they are not the ideal methods for this type of image, and the results of binarization show that the foreground and background are mixed together. Although the PFE method has a poor segmentation effect on text images, it performs well on nontext images, and some results are even better than our method, such as row 2 of Fig 24c. Our method shows its robustness, as shown in Fig 24d.

Based on all experiments, we calculated the evaluation indicators for the six comparison methods and our method, which were the average results of all experiments. The final quantitative evaluations are shown in Table 3, which illustrates that compared with these methods, our method has advantages except for time. Then, we compared our method only to Long's method.

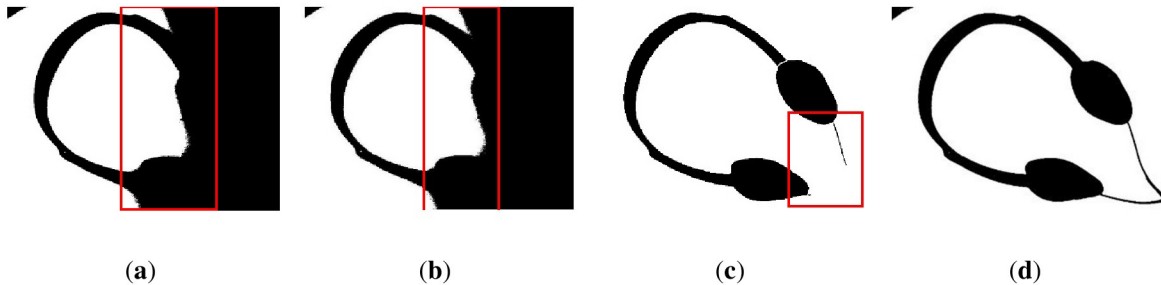

|     (a)     |     (b)     |     (c)     |     (d)     |

**Fig 22. The results of Fig 17a.** (**a**) Otsu's method, (**b**) k-means method, (**c**) PFE method, and (**d**) proposed method. Some flaws of the comparison methods are shown in red frames.

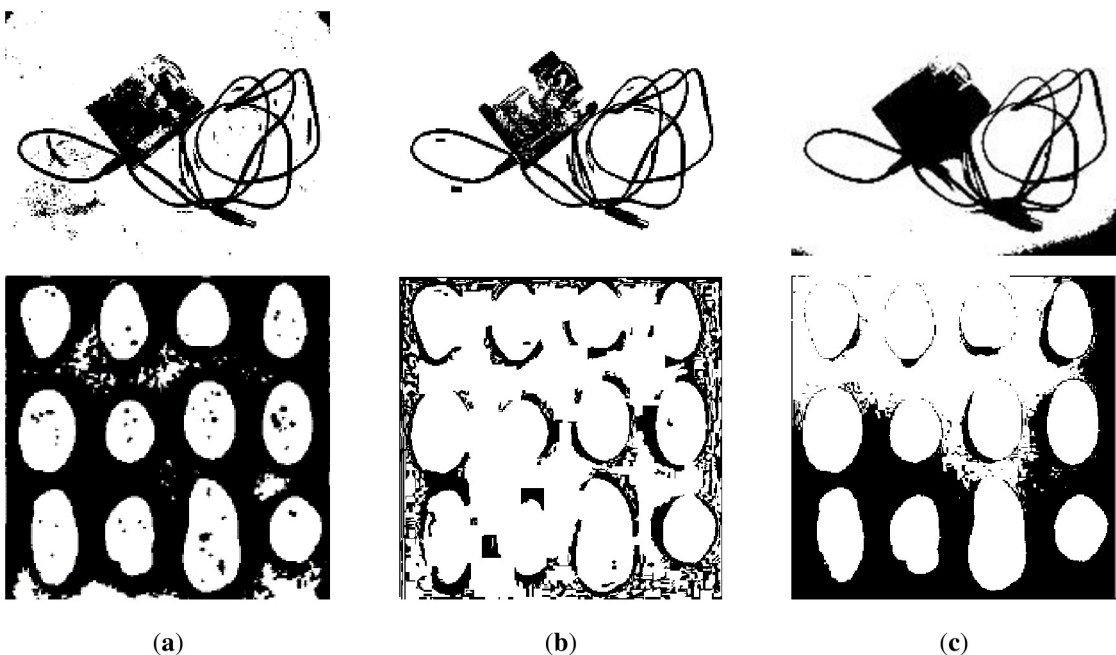

**Fig 23. The results of some nontext images.** (**a**) water flow method, (**b**) SSP method, and (**c**) Wang's method.

Regarding the previous content, because our method expands on Long's method [17], the comparison results and tables of the two are separately listed to show that our method is more advanced. The parameters of our method are $\alpha = 15$, $\beta = 1.2$, and $\gamma = 0.3$. The parameters of Long's method are $\sigma = 20$ and $\gamma = 0.3$. It can be seen from Table 4 that the difference in the average performance is not prominent, but the method in this paper requires less time than

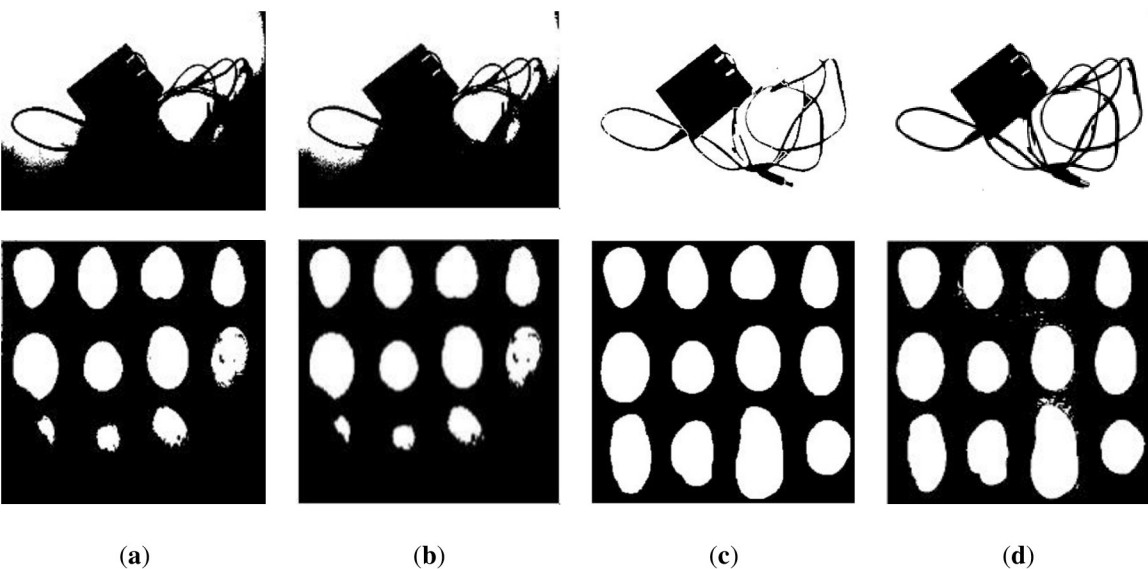

**Fig 24. The results of some nontext images.** (**a**) Otsu's method, (**b**) k-means method, (**c**) PFE method, and (**d**) proposed method.

**Table 3. Quantitative performance of the six comparison methods and our method.**

| Method | ME | MIOU | PA | F_Measure | Time (ms) |
|---|---|---|---|---|---|
| Water flow | 0.1379 | 0.7661 | 0.8621 | 0.8725 | 516 |
| SSP | 0.0953 | 0.4536 | 0.9047 | 0.9493 | 967 |
| Wang's | 0.2101 | 0.5891 | 0.7899 | 0.8472 | 770 |
| Otsu | 0.3151 | 0.5103 | 0.6849 | 0.7114 | **1** |
| K-means | 0.4241 | 0.3450 | 0.5758 | 0.5575 | 667 |
| PFE | 0.0571 | 0.8496 | 0.9428 | 0.9460 | - |
| Ours | **0.0369** | **0.9112** | **0.9636** | **0.9697** | 524 |

**Table 4. Quantitative performance of Long's method and our method.**

| Method | ME | MIOU | PA | F_Measure | Time (ms) |
|---|---|---|---|---|---|
| Long's | **0.0131** | 0.9426 | 0.9710 | 0.9693 | 828 |
| Ours | 0.0298 | **0.9445** | **0.9801** | **0.9782** | **510** |

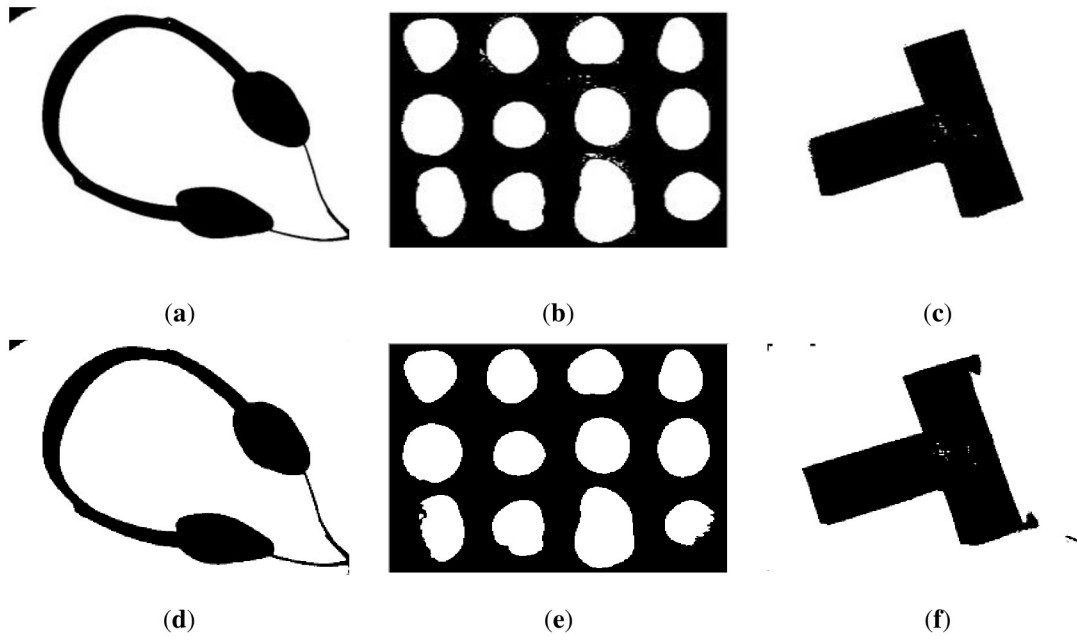

(a)  (b)  (c)

(d)  (e)  (f)

**Fig 25. Demonstration of our method (a)~(c) and Longs' method (d)~(f).**

Long's method. The visual displays are shown in Fig 25, which shows that our method is better for the binary processing of details.

## Conclusion

The method introduced in this paper, which combines the GSS with spectrum decomposition to estimate background, is mainly used for the background estimation of unevenly illuminated images. The main target (foreground) is extracted by using the difference between the background and the original image. Our method can use various thresholding segmentation

algorithms for the final binarization results. In this paper, we choose some effective thresholding methods, such as Otsu's method.

After considering a large number of experiments and comparing a variety of segmentation methods, the results show that our method can effectively threshold images with unevenly illuminated backgrounds. However, the method in this paper has some drawbacks in text image segmentation in scenes with particularly bright light, which can be overcome by improving the Gaussian scale space. Therefore, we can propose the following improvements for our method. First, we can improve the construction of the GSS. Then, we can use other dimension reduction algorithms. Furthermore, we can also apply a postprocessing method to further improve the results.

## Supporting information

**S1 Dataset.**
(RAR)

## Author Contributions

**Conceptualization:** JianWu Long.

**Data curation:** JianWu Long.

**Formal analysis:** ZeRan Yan.

**Software:** ZeRan Yan.

**Supervision:** XinLei Song.

**Validation:** HongFa Chen.

**Writing – original draft:** ZeRan Yan.

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
