## [Decision Letter · Decision Letter 0]

2 Dec 2020

PONE-D-20-30273

Spectrum Decomposition in Gaussian Scale Space for Uneven-illumination Image Binarization

PLOS ONE

Dear Dr. ran,

Thank you for submitting your manuscript to PLOS ONE. After careful consideration, we feel that it has merit but does not fully meet PLOS ONE’s publication criteria as it currently stands. Therefore, we invite you to submit a revised version of the manuscript that addresses the points raised during the review process.

We look forward to receiving your revised manuscript.

Kind regards,

Seyedali Mirjalili

Academic Editor

PLOS ONE

Journal Requirements:

"This work was supported by the Humanities and Social Sciences Research Key Program of Chongqing Municipal

Education Commission (Grant No. 17SKG136), the National Natural Science Foundation of China for Young

Scientists (Grant No. 61502065) and the Foundation and Frontier Research Key Program of Chongqing Science

and Technology Commission (Grant No. cstc2015jcyjBX0127)."

4. Please ensure that you refer to Figure 14 in your text as, if accepted, production will need this reference to link the reader to the figure.

5. We note you have included tables to which you do not refer in the text of your manuscript. Please ensure that you refer to Tables 2 and 4 in your text; if accepted, production will need this reference to link the reader to the Table.

6. We note in your Data Availability statement you have advised "No - some restrictions will apply".

Could you please  clarify the nature of these restrictions, ie. If due to ethical or legal reasons.

PLOS defines a study's minimal data set as the underlying data used to reach the conclusions drawn in the manuscript and any additional data required to replicate the reported study findings in their entirety. All PLOS journals require that the minimal data set be made fully available. For more information about our data policy, please see http://journals.plos.org/plosone/s/data-availability.

Reviewers' comments:

Reviewer's Responses to Questions

**Comments to the Author**

1. Is the manuscript technically sound, and do the data support the conclusions?

Reviewer #1: Yes

Reviewer #2: Yes

2. Has the statistical analysis been performed appropriately and rigorously? 

Reviewer #1: Yes

Reviewer #2: Yes

3. Have the authors made all data underlying the findings in their manuscript fully available?

Reviewer #1: Yes

Reviewer #2: Yes

4. Is the manuscript presented in an intelligible fashion and written in standard English?

Reviewer #1: Yes

Reviewer #2: Yes

5. Review Comments to the Author

Reviewer #1: The authors proposed an extend work of long [15] which is based on a Gaussian space method

to segment non-illumination text image, the paper is well written and presented in good manner however:

1- The related works/ literature is poor. It is recommended to include recent studies on the same domain to further enhance the presentation of the paper. Add more references.

2- Fig 2. Flowchart of all processes, it is recommended to improve the presentation of this flowchart, make it clear because it represents the whole processes of your method.

3- Fig 7 and 8, consists of parts from a-h, it is recommended to explain each graph (a) represents what? b represents what ?and so on...

Reviewer #2: 1. References used in the paper are not latest. Include some paper from year 2019 and 2020.

2. Authors compared their proposed method with the conventional techniques like: Otsu method, k means etc. Compare the proposed work with edge based and deep learning based techniques.

6. PLOS authors have the option to publish the peer review history of their article (what does this mean?). If published, this will include your full peer review and any attached files.

Reviewer #1: No

Reviewer #2: No

---

## [Author Response · Author response to Decision Letter 0]

2 Jan 2021

Replies to the reviewers’ comments.

Reviewer #1:

1. The related works/ literature is poor.

Response: Introduction has added some literatures about segmentation area from 2019 to 2020, which include one survey, and some recently traditional methods, deep learning methods. The survey can give us a lot of references. The Deep learning method use multi-scale and multi-level architecture, this structure can extract more information and feature from images. 

2. Make the Fig 2 (Flowchart of all processes) clear.

Response: This has been revised.

3. Fig 7 and 8, consists of parts from a-h, it is recommended to explain each graph (a) represents what? b represents what?

Response: I’m sorry we did not clarify the intention of Fig 7 and 8. The two figures They represent two type of testing text images. Because of our experiments show that different methods have different sensitivity to images under different environmental brightness. In other words, some methods have better performance in brighter image while some or not. The legends of Fig 7 and Fig have related description. Therefore, we divide the images as brighter data and darker data. 

In addition, we did not add the representation of a-h om Fig 7 and 8. Because the data are text images, all of them have rich borders and textures while having uneven lighting background.

Reviewer #2:

1. References used in the paper are not latest.

Response: Sorry for not being able to show recent related research in the article. To introduce the work in the field of image segmentation in recent years, the references have added some literatures about segmentation area from 2019 to 2020, which include one survey, and some recently traditional methods, deep learning methods. The survey can give us a lot of references. The Deep learning method use multi-scale and multi-level architecture, this structure can extract more information and feature from images. One of the newly added traditional methods is used for comparison.

2. Compare the proposed work with edge based and deep learning based techniques.

Response: Thank you for your suggestions, and we compared two edge-based methods, BSR [1] method and PFE [2] method. However, we just demonstrate the PFE in our paper, the performance of BSR is relatively poor on both text images and non-text images.The segmentation of PFE on text images is not satisfactory, but it does a good performance of segmentation on non-text images. For the PFE method, we first use the deep learning-based boundary detector HED [3] recommended by the PFE author’s Github to detect the boundary. However, if detected boundary is not good, segmentation tend to perform poor.

As for the deep learning method, because our method is designed to segment images with unevenly illuminated under industrial conditions with a small amount of data, it is easy to overfit our data by using a deep network when the amount of test data is small. Therefore, we did not use deep learning methods for comparison.

References

1. P. Arbeláez, M. Maire, C. Fowlkes and J. Malik, "Contour Detection and Hierarchical Image Segmentation," in IEEE Transactions on Pattern Analysis and Machine Intelligence, vol. 33, no. 5, pp. 898-916, May 2011, doi: 10.1109/TPAMI.2010.161. 

2. Fang C, Liao Z, Yu Y. Piecewise Flat Embedding for Image Segmentation. IEEE Transactions on Pattern Analysis and Machine Intelligence. 2019;41(6):1470–1485.

3. Xie S, Tu Z. Holistically-nested edge detection[C]//Proceedings of the IEEE international conference on computer vision. 2015: 1395-1403.

---

## [Decision Letter · Decision Letter 1]

25 Mar 2021

PONE-D-20-30273R1

Spectrum decomposition in gaussian scale space for uneven illumination image binarization

PLOS ONE

Dear Dr. ran,

Thank you for submitting your manuscript to PLOS ONE. After careful consideration, we feel that it has merit but does not fully meet PLOS ONE’s publication criteria as it currently stands. Therefore, we invite you to submit a revised version of the manuscript that addresses the points raised during the review process.

We look forward to receiving your revised manuscript.

Kind regards,

Saeed Mian Qaisar, Ph.D.

Academic Editor

PLOS ONE

Journal Requirements:

Additional Editor Comments (if provided):

Dear authors, please submit a "response to reviewers comments" while clearly addressing all raised points. Also carefully revise the manuscript in order to improve its language and clarity. If you are prepared to undertake the required work, I would be pleased to reconsider my decision.

Reviewers' comments:

Reviewer's Responses to Questions

2. Is the manuscript technically sound, and do the data support the conclusions?

Reviewer #1: Yes

3. Has the statistical analysis been performed appropriately and rigorously? 

Reviewer #1: Yes

4. Have the authors made all data underlying the findings in their manuscript fully available?

Reviewer #1: Yes

5. Is the manuscript presented in an intelligible fashion and written in standard English?

Reviewer #1: Yes

7. PLOS authors have the option to publish the peer review history of their article (what does this mean?). If published, this will include your full peer review and any attached files.

Reviewer #1: No

---

## [Author Response · Author response to Decision Letter 1]

14 Apr 2021

Reviewer #1:

1. The related works/ literature is poor.

Response: Introduction has added some literatures about segmentation area from 2019 to 2020, which include one survey, and some recently traditional methods, deep learning methods. The survey can give us a lot of references. The Deep learning method use multi-scale and multi-level architecture, this structure can extract more information and feature from images. 

2. Make the Fig 2 (Flowchart of all processes) clear.

Response: This has been revised.

3. Fig 7 and 8, consists of parts from a-h, it is recommended to explain each graph (a) represents what? b represents what?

Response: I’m sorry we did not clarify the intention of Fig 7 and 8. The two figures They represent two type of testing text images. Because of our experiments show that different methods have different sensitivity to images under different environmental brightness. In other words, some methods have better performance in brighter image while some or not. The legends of Fig 7 and Fig have related description. Therefore, we divide the images as brighter data and darker data. 

In addition, we did not add the representation of a-h om Fig 7 and 8. Because the data are text images, all of them have rich borders and textures while having uneven lighting background.

Reviewer #2:

1. References used in the paper are not latest.

Response: Sorry for not being able to show recent related research in the article. To introduce the work in the field of image segmentation in recent years, the references have added some literatures about segmentation area from 2019 to 2020, which include one survey, and some recently traditional methods, deep learning methods. The survey can give us a lot of references. The Deep learning method use multi-scale and multi-level architecture, this structure can extract more information and feature from images. One of the newly added traditional methods is used for comparison.

2. Compare the proposed work with edge based and deep learning based techniques.

Response: Thank you for your suggestions, and we compared two edge-based methods, BSR [1] method and PFE [2] method. However, we just demonstrate the PFE in our paper, the performance of BSR is relatively poor on both text images and non-text images, we demonstrate some results in following Fig 1. 

The segmentation of PFE on text images is not satisfactory, but it does a good performance of segmentation on non-text images. For the PFE method, we first use the deep learning-based boundary detector HED [3] recommended by the PFE author’s Github to detect the boundary. However, if detected boundary is not good, segmentation tend to perform poor, which are showed by Fig 2.

As for the deep learning method, because our method is designed to segment images with unevenly illuminated under industrial conditions with a small amount of data, it is easy to overfit our data by using a deep network when the amount of test data is small. Therefore, we did not use deep learning methods for comparison.

References

1. P. Arbeláez, M. Maire, C. Fowlkes and J. Malik, "Contour Detection and Hierarchical Image Segmentation," in IEEE Transactions on Pattern Analysis and Machine Intelligence, vol. 33, no. 5, pp. 898-916, May 2011, doi: 10.1109/TPAMI.2010.161. 

2. Fang C, Liao Z, Yu Y. Piecewise Flat Embedding for Image Segmentation. IEEE Transactions on Pattern Analysis and Machine Intelligence. 2019;41(6):1470–1485.

3. Xie S, Tu Z. Holistically-nested edge detection[C]//Proceedings of the IEEE international conference on computer vision. 2015: 1395-1403.

---

## [Editor Report · Decision Letter 2]

19 Apr 2021

Spectrum decomposition in gaussian scale space for uneven illumination image binarization

PONE-D-20-30273R2

Dear Dr. ran,

We’re pleased to inform you that your manuscript has been judged scientifically suitable for publication and will be formally accepted for publication once it meets all outstanding technical requirements.

Kind regards,

Saeed Mian Qaisar, Ph.D.

Academic Editor

PLOS ONE

Additional Editor Comments (optional):

Dear authors thank you very much for submitting your paper to the PLOS ONE journal. I am pleased to inform you that your revised paper has been accepted for publication.
---

## [Editor Report · Acceptance letter]

21 Apr 2021

PONE-D-20-30273R2 

Spectrum decomposition in gaussian scale space for uneven illumination image binarization 

Dear Dr. Yan:

I'm pleased to inform you that your manuscript has been deemed suitable for publication in PLOS ONE. Congratulations! Your manuscript is now with our production department. 

Kind regards, 

on behalf of

Dr. Saeed Mian Qaisar 

Academic Editor

PLOS ONE